

# The first observations of *Ischnochiton* (Mollusca, Polyplacophora) movement behaviour, with comparison between habitats differing in complexity

Kiran Liversage[1] and Kirsten Benkendorff[2]

[1] Estonian Marine Institute, University of Tartu, Tallinn, Estonia
[2] Marine Ecology Research Centre, Southern Cross University, Lismore, NSW, Australia

Corresponding author
Kiran Liversage,
kiran.liversage@ut.ee

## ABSTRACT

Most species of *Ischnochiton* are habitat specialists and are almost always found underneath unstable marine hard-substrata such as boulders. The difficulty of experimenting on these chitons without causing disturbance means little is known about their ecology despite their importance as a group that often contributes greatly to coastal species diversity. In the present study we measured among-boulder distributional patterns of *Ischnochiton smaragdinus*, and used time-lapse photography to quantify movement behaviours within different habitat types (pebble substrata and rock-platform). In intertidal rock-pools in South Australia, *I. smaragdinus* were significantly overdispersed among boulders, as most boulders had few individuals but a small proportion harboured large populations. *I. smaragdinus* individuals emerge from underneath boulders during nocturnal low-tides and move amongst the inter-boulder matrix (pebbles or rock-platform). Seventy-two percent of chitons in the pebble matrix did not move from one pebble to another within the periods of observation (55–130 min) but a small proportion moved across as many as five pebbles per hour, indicating a capacity for adults to migrate among disconnected habitat patches. Chitons moved faster and movement paths were less tortuous across rock-platform compared to pebble substrata, which included more discontinuities among substratum patches. Overall, we show that patterns of distribution at the boulder-scale, such as the observed overdispersion, must be set largely by active dispersal of adults across the substratum, and that differing substratum-types may affect the degree of adult dispersal for this and possibly other under-boulder chiton species.

## INTRODUCTION

Dispersal of mobile benthic species can occur by a combination of movement processes occurring as adults (*Little, 1989*; *Grantham, Eckert & Shanks, 2003*) and by 'supply-side' processes (*Underwood & Fairweather, 1989*) for species with larval stages. Contributions of adult and larval processes to dispersal have been measured for species with easily observable larval processes (e.g. larval settlement/recruitment; *Rowley, 1989*) or adult

processes (e.g. movement of slow-moving species on rock-platforms exposed during low tide; *Underwood & Chapman, 1989*). There are many species, however, that occur almost exclusively in cryptic/hidden habitats that are not easily observed, such as underneath unstable hard-substrata. The species are mostly hidden from view, and in order to observe them it is generally required to disturb the habitat (*Chapman & Underwood, 1996*). Consequently, we have little information about their natural dispersal capacities as adults. Our knowledge about intertidal invertebrate behavioural ecology (see reviews by *Grantham, Eckert & Shanks, 2003*; *Ng et al., 2013*) would be improved by incorporating under-boulder species, because the specialist species there often have high levels of rarity or endemism (*Benkendorff & Przeslawski, 2008*; *Chapman, Underwood & Clarke, 2009*; *Liversage, 2015*), and ecological information is needed to inform conservation management.

One of the most widespread groups of boulder habitat specialists are chitons within the *Ischnochiton* genus. While much research has focused on movement behaviours of other chitons that live on exposed (i.e. non-cryptic) rocky habitats (*Thorne, 1968*; *Chelazzi, Focardi & Deneubourg, 1983*, *1988*; *Chelazzi, Della Santina & Parpagnoli, 1987*) practically nothing is known of natural movement behaviours of *Ischnochiton* that are primarily associated with boulders. *Palmer (2012)* suggested that patterns of among-boulder overdispersion could be explained by philopatric behaviour, with chitons rarely dispersing from their natal boulders. *Chapman (2002)* observed high rates of dispersal onto artificially deployed boulders, and questioned whether such dispersal may occur by 'drifting' or 'crawling.' *Smith & Otway (1997)* and *Jörger, Meyer & Wehrtmann (2008)* noted that some chitons readily drop off overturned boulders and fall into the water to be passively transported by water motion. This was considered an 'escape-response' that may affect the movements and distribution of species that use this behaviour. Empirical data about movement ecology of *Ischnochiton* is required to determine whether dispersal behaviours such as these are occurring in reality and contributing to distributional patterns such as overdispersion. This pattern occurs when large variation among replicates causes data to not approximate a Poisson distribution (*Richards, 2008*) and has been observed repeatedly for distribution data of other *Ischnochiton* species (*Grayson & Chapman, 2004*; *Liversage & Benkendorff, 2013*).

The only direct observations of *Ischnochiton* movements are notes of movement occurring on exposed rock surfaces, mostly nocturnally (*Kangas & Shepherd, 1984*). Using shell patterns to identify individual chitons, *Liversage et al. (2012)* found that approximately two-thirds of the individuals emigrate from their original boulder over three days, while an average of three new chiton individuals move onto boulders. Similarly, in an intertidal cobble reef, *McClintock, Angus & McClintock (2007)* marked and relocated a habitat-generalist chiton species (*Sypharochiton pelliserpentis*) and found that the percentage of chitons that stayed under their original cobble after two tidal cycles varied from 10% (small cobbles) to 45% (large cobbles). These studies suggest chitons may move frequently across boulder habitat patches. The boulders/cobbles in these studies were, however, overturned for sampling and hence physically disturbed (*Chapman & Underwood, 1996*), so the relatively high levels of movement recorded may

not fully reflect natural movement patterns. To our knowledge no previous study has quantified undisturbed movement patterns of any *Ischnochiton* species, which was our aim in this study.

Temperate boulder reefs of the south-eastern Australian intertidal-zone harbour populations of *Ischnochiton smaragdinus*. This small chiton attains a maximum length of approximately 2 cm and has a carnivorous diet of sponges, bryozoans and ascidians (*Kangas & Shepherd, 1984*). One trait of this species that is atypical among most *Ischnochiton* is that some individuals become active on the upper-surfaces of exposed-rock habitats nocturnally. Some other species of under-boulder chitons that are more rare also share this behaviour (*Kangas & Shepherd, 1984*), but for many parts of Australia, *I. smargdinus* is the only species that can be observed in large abundances exhibiting this behaviour.

The present study focused on *I. smaragdinus*, a common chiton in South Australian rock-pools which contain a variety of substrata including mostly pebbles (diameter 4–64 mm), boulders (diameter >256 mm) and rock-platforms. First, we measured distributional patterns to test the hypothesis that adult individuals are overdispersed among boulders, similarly to many other representatives of *Ischnochiton* (*Grayson & Chapman, 2004*; *Liversage & Benkendorff, 2013*). These distributions may be caused by movement behaviours. Accordingly, we measured movement paths of chitons using time-lapse photography during nocturnal low tides. To determine the generality of the finding from *Liversage et al. (2012)*, that chitons migrate among habitat patches that have been disturbed by sampling, we tested the hypothesis that chitons will not remain on individual habitat patches, but migrate amongst the boulders or pebbles that have been left undisturbed (i.e. not overturned or moved). Measurements of chiton movements were also made within pools that included rock-platform habitat, which includes fewer discontinuities (i.e. interstices and areas of sand between adjacent hard-substrata) among habitat patches compared to rock-pools containing pebbles. We tested the hypothesis that variables including speed and directionality were affected by the different habitat types.

## MATERIALS AND METHODS

Distributions and movement patterns of *I. smaragdinus* were measured during low tides at four sheltered intertidal boulder fields on the Fleurieu Peninsula, South Australia during daylight. For distribution measurements, 30 haphazardly selected boulders were overturned and numbers of attached *I. smaragdinus* were counted during October 2007 at two sites (Myponga Beach—35°22′12.6″S 138°23′18.2″E and Second Valley—35°30′39.6″S 138°12′58.4″E). The boulders at both locations were approximately 30 cm long and 15 cm high, and were siltstone or sandstone. Sampled boulders were separated from each other by approximately 1 m. Frequency distributions of chitons across boulders at each site were compared to a Poisson distribution (expected if chitons are distributed randomly) using a one-sample Kolmogorov–Smirnov test. If patterns of overdispersion produce non-random frequency distributions, this test will indicate a significant difference between observed and expected (Poisson) distributions.

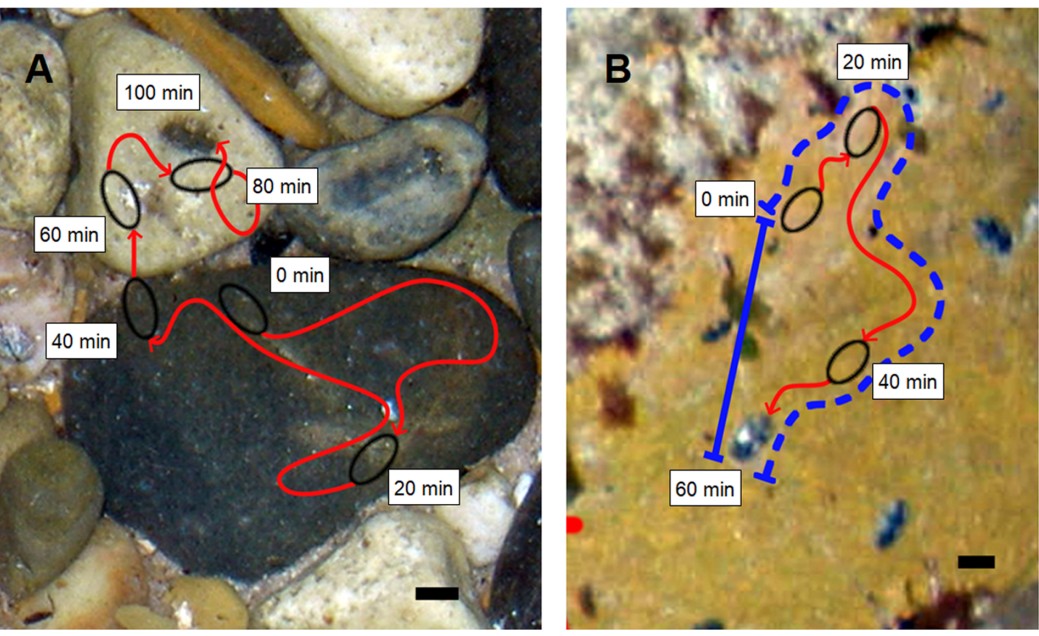

**Figure 1  Representative movement paths of *Ischnochiton smaragdinus*.** Movement paths are from (A) pebble substratum (100 min) and (B) rock-platform substratum (60 min) that were photographed from above the water surface (depth approx. 20–30 cm). The dashed blue line shown in (B) represents the Gross Displacement and the solid blue line indicates Net Displacement, with the ratio of these measures calculated to indicate the tortuosity of the movement path. Black lines at bottom right = 1 cm.

   Movement paths of *I. smaragdinus* were measured using time-lapse photography techniques involving photography from above the water surface in randomly selected rock-pools. Because no individuals were observed active on upper-surfaces of substrata during daytime, the movement paths were measured at night, which was done during seven low-tide periods at two sites at Myponga Beach between August and December 2006. One site (35°22′12.6″S 138°23′18.2″E) had rock-pools with pebble substrata (Fig. 1A); the mean length of pebbles was measured from four rock-pools to be 4.14 cm (SE = 0.35) and all measured rock-pools had pebbles of similar length (ANOVA $F_{(3, 36)} =$ 0.88, $P > 0.25$). The pebbles were flattened with their height about half their length, and were partially buried in sand. The second site (35°22′01.9″S 138°24′19.1″E) had rock-pools with a substratum of unbroken rock-platform (Fig. 1B). It is unknown if temporal effects may have occurred across the time the sites were sampled, but the two habitat types were sampled in alternation and during similar weather conditions to avoid confounding. Some nights were during full moon and others new moon phases, but a preliminary comparison (ANOVA) done at one site during these two conditions found no significant differences ($P > 0.05$) in any movement variables (see below for description of variables). At each rock-pool, a camera (Olympus C5050; Olympus, Shinjuku, Tokyo, Japan) was positioned on a tripod directly above to photograph an area of approximately 50 × 50 cm. The camera took digital images at 1 min intervals using the flash set on minimum power. Flash photography did not appear to affect the behaviour of the chitons, e.g. the distances moved immediately following the first exposure to the flash were similar to those after

30 min of further exposures (ANOVA $F_{(1, 62)} = 2.20$, $P > 0.1$) and the numbers of chitons active outside of refuges also did not change between those times (ANOVA $F_{(1, 13)} = 0.04$, $P > 0.75$). Periods of photography were initiated when the tide had receded and there were no visual distortions of the substratum from moving water. Observations were ended when distortion from increasing water depth during the advancing tide again prevented resolution of the substratum. This method provided an observation time that varied from 55 to 130 min.

Images were processed using Photoshop CS imaging software (Adobe, San Jose, CA, USA). The locations of all chitons within the rock-pool in the first image in a given time series was marked on a Photoshop workspace layer separate from the image, then the next image in the time series was superimposed on the first. The next image was adjusted in scale to offset the effects of changes in water level (which magnifies or reduces the view of the substratum when viewed above the water surface), and the subsequent position of all chitons was then added to the layer onto which the initial positions were marked. This process was used for all subsequent images in the time series until the entire movement paths of all chitons were delineated. To ensure each chiton followed was an independent replicate, they were only included if no interaction occurred with other chitons (i.e. direct contact or movement across another chiton's path). Studies using photographs to measure habitat characteristics on larger boulder habitats have used techniques to correct for the curvature of boulders (e.g. applying a corrective factor of ≈1.2 to area measurements from boulder edges; *Liversage et al., 2012*). While no distortion would be caused when delineating movement paths leading around edges of pebbles, there may be distortion of paths leading from edges over the pebble tops. However, because the pebbles were flattened in shape, and the chitons were large (mean 0.64 cm) in relation to the pebbles (mean 4.14 cm), it was not considered necessary in this case to apply any corrective adjustments.

Three movement variables were measured as well as the length of each chiton. Net speed was calculated as the total distance travelled divided by the total time observed. Maximum speed was considered as the largest distance travelled by an individual in any 5 min period. We calculated the net:gross displacement ratio to provide an indication of the tortuosity of movement paths, as done for sea stars (*Swenson & McClintock, 1998*), copepods (*Buskey, 1984*), fish (*Parrish, Viscido & Grünbaum, 2002*) and seals (*Davis et al., 2001*). This metric is calculated as net displacement (the straight distance between the start and end point of a movement path) divided by gross displacement (the actual distance travelled; Fig. 1). A net:gross displacement ratio close to 0 indicates the movement path is highly tortuous, and most movement has not contributed to dispersal away from the starting point. A value close to 1 indicates the path is straighter with most movement having resulted in dispersal. The net:gross displacement ratio was measured only for chitons that moved a minimum of 30 mm during the period of observation.

The number of chitons in each rock-pool was variable, so comparisons among treatments were done with univariate PERMANOVA, using PRIMER v6 (*Anderson & Walsh, 2013*). We considered substratum-type as a fixed factor and the different

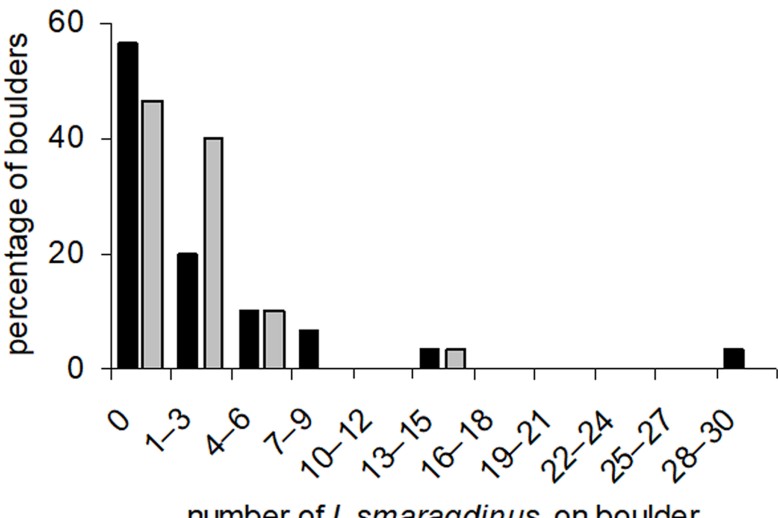

**Figure 2 Frequency distributions of *Ischnochiton smaragdinus* among boulders.** Percentages are shown of the numbers of boulders that harboured different numbers of *Ischnochiton smaragdinus* individuals, from the sites Myponga Beach and Second Valley (30 boulders sampled per site).

rock-pools as a random factor. There was only one substratum type in each of the rock-pools, so the rock-pool factor was nested in substratum-type. Attempts were made to find rock-pools with different substratum-types within the same area, but appropriate substratum-types were only found separated spatially. Distributional patterns vary over small spatial scales (i.e. among boulders) for *Ischnochiton* (*Grayson & Chapman, 2004*; *Liversage & Benkendorff, 2013*), but few differences in other among-boulder movement patterns have been found during comparisons of separate larger-scale locations (*Liversage et al., 2012*). This suggests that spatial confounding between the sites can be considered unlikely in our comparisons between these substratum-types. The analyses used Euclidean distances and 9,999 permutations. Homogeneity of variances was tested with PERMDISP, using medians, which in univariate analyses is equivalent to Levene's test (*Anderson, Gorley & Clarke, 2008*). If the *P* values of the random factor was >0.25, it was eliminated to provide a more powerful test for the relevant null hypothesis (*Underwood, 1997*).

## RESULTS

At our two study sites, most boulders had no *I. smaragdinus* or only a small number (one–three individuals; Fig. 2) living under them. A small percentage of boulders harboured many individuals, reaching up to 30 individuals at Myponga beach and 13 at Second Valley (Fig. 2). These right-skewed frequency distributions differed strongly from the Poisson distributions expected if chitons were distributed randomly (Myponga Beach Kolmogorov–Smirnov goodness-of-fit test $P < 0.001$; Second Valley $P < 0.001$). These

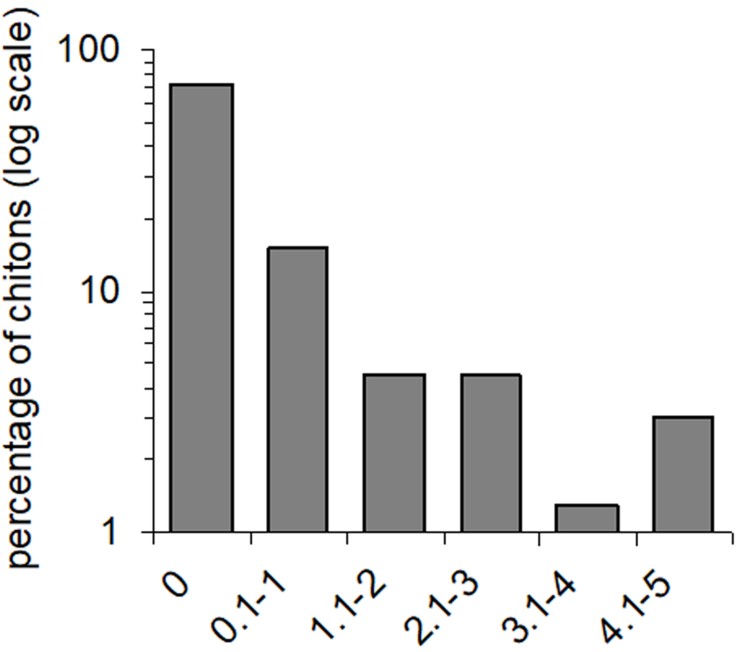

**Figure 3 Amounts of movement across pebble habitat patches.** Percentages of *Ischnochiton smaragdinus* from a total of 82 individuals that had different rates of movement across pebbles in four seperate rock-pools with a pebble substratum.

measurements were taken diurnally during low tide, and all 143 individuals were sheltering underneath the boulders. None occurred in exposed areas that were visible without overturning the substrata.

Movement paths of 113 individuals were analysed. Only two individuals did not move during the observation periods (although they had likely moved recently into the exposed habitat). The frequency of movement generally did not result in dispersal across substratum units (pebbles), but a smaller proportion of individuals displayed more extensive dispersal (Fig. 3). Substratum-type appeared to affect all movement variables measured (Table 1). Significantly lower speeds were observed in rock-pools with a pebble substratum, and this difference was particularly evident regarding the maximum speeds attained (Fig. 4A). The fastest individual overall speed over a 5 min period was 55 cm h$^{-1}$ on a rock-platform surface. Although mean sizes of chitons varied among random rock-pools, they did not vary between the two habitat types (Table 1), so this variable did not affect differences in speed between fixed factor treatments.

The net:gross displacement ratio was significantly greater in rock-pools with a rock-platform substratum (Table 1; Fig. 4B), indicating more directional, less tortuous movement paths. Although chitons on pebble substrata often had highly tortuous paths, in no instances was it observed that a given chiton returned to the same position where it had previously been resting.

**Table 1 Univariate PERMANOVA comparing the length, movement speed, and tortuosity of movement paths for *Ischnochiton smaragdinus* individuals on a pebble or rock-platform substratum.**

| Source | Chiton length | | | Net speed | | | Maximum speed | | | Net: gross displacement ratio | | |
|---|---|---|---|---|---|---|---|---|---|---|---|---|
| | *df* | MS | F | *df* | MS | F | *df* | MS | F | *df* | MS | F |
| Substratum-type | 1 | 0.001 | 0.11 | 1 | 3937 | 6.12** | 1 | 121.44 | 9.28** | 1 | 0.412 | 7.978* |
| Rock-pool (nested) | 5 | 0.130 | 2.84* | 5 | – | | 5 | – | | 5 | 0.007 | 2.304 |
| Residual | 106 | 4.687 | | 112 | 643 | | 112 | 13.08 | | 80 | 0.003 | |
| | PERMDISP *P* > 0.75 | | | PERMDISP *P* > 0.1 | | | PERMDISP *P* > 0.1 | | | PERMDISP *P* > 0.1 | | |

**Notes:**
Measurements were taken from seven randomly selected rock-pools. Substratum-type was a fixed factor and rock-pool was random and nested. PERMDISP tests determined if variances were significantly heterogenous. When the *P*-value of the random factor was >0.25 it was eliminated from the analysis to provide a more powerful test for the relevant null hypothesis (*Underwood, 1997*). '–' designates eliminated term.
* *P* < 0.05.
** *P* < 0.01.

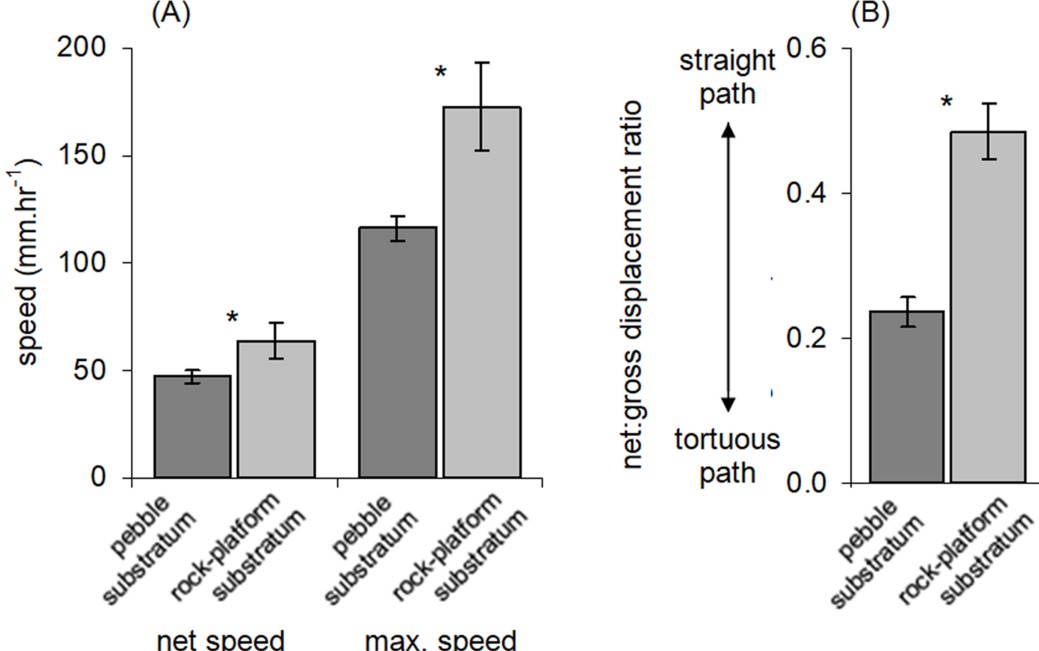

**Figure 4 Movement characteristics of *Ischnochiton smaragdinus* in habitats of differing complexity.** Mean (±SE) (A) speed of movement and (B) tortuosity of the movement path (measured as net:gross displacement ratio; see Fig. 1). 'Net speed' refers to the speed averaged over the entire observation period (55–130 min) and 'max. speed' refers to fastest rate of movement by a chiton in any five minute period. Data are from four rock-pools with pebble substratum (81 chitons measured) and three with rock-platform substratum (32 chitons). *P* < 0.05.

## DISCUSSION

Movement patterns of mobile intertidal species vary from almost no movement (*Branch, 1975*), to movement to and from a 'home' position (*Mackay & Underwood, 1977*; *Chelazzi, 1990*), to regular and widespread dispersal among habitat patches (*Underwood, 1977*). The results from the present study show for the first time that the undisturbed movement behaviour of *I. smaragdinus* involves dispersal among multiple patches of substrata. Many individuals did not move across pebbles during the observation periods,

but a small proportion moved across as many as five within an hour. These individuals clearly did not remain in their natal habitat, thus such behaviour is unlikely to explain patterns of overdispersion in this species. No instances of chitons being 'drifted' in the water were observed, indicating that adult dispersal occurs via 'crawling,' at least during low tide (*Chapman, 2002*). This species does not exhibit the behaviour of curling and detaching from disturbed boulders that other *Ischnochiton* do (*Smith & Otway, 1997*; *Jörger, Meyer & Wehrtmann, 2008*), and for these other species it is possible that more migration via drifting occurs. *I. smaragdinus* may have migration behaviour similar to that of the intertidal limpet *Cellana tramoscerica* which, like many chitons (*Thorne, 1968*; *Chelazzi, Focardi & Deneubourg, 1988*), sometimes displays homing behaviour, and it alternates between the homing behaviour and randomly directed movement (*Mackay & Underwood, 1977*). The observations here could be explained by *I. smaragdinus* individuals remaining in a 'home' patch (i.e. pebble) for a certain time and then subsequently moving quickly though adjacent patches. The methods in the movement observations only assessed chitons that were active and many more individuals may have been sheltering underneath the boulders and pebbles that were not observed, so the extent of population-level migration or patch fidelity is unknown. It is also unknown how this species may be moving during other contexts. For example, other forms of movement may be occurring during day- or night-time high tides (although no individuals were observed moving during daylight low tides) and for populations living deeper in the subtidal zone (*Kangas & Shepherd, 1984*).

Similar to other intertidal molluscs (*Underwood & Chapman, 1989*; *Chapman & Underwood, 1994*; *Erlandsson, Kostylev & Williams, 1999*; *Underwood, 2004*), movements of *I. smaragdinus* appear to be affected by topography of the substratum. A discontinuous layer of pebbles was associated with reduced speed of movement and resulted in more convoluted movement paths. Complex topographies are known to reduce movement speeds of some gastropods and result in faster population turn-over rates in less complex areas with greater immigration and emigration (*Underwood & Chapman, 1989*). It may be advantageous for chitons to minimise time spent on exposed rock-platforms to avoid predation, especially from brachyurans known to prey on *small chitons* (*Mendonça et al., 2016*), although brachyurans in South Australian rock-pools such as *Ozius truncatus* may still be able to prey on chitons underneath pebbles by overturning them, which was observed on one occasion in the time-lapse photographs of this study.

The differences in dispersal capacity between the habitat types may also be useful for understanding processes involving disturbance and restoration ecology. Disturbance in the form of movement or overturning of boulders, or disruption of the under-boulder substratum, reduces densities of chitons before a subsequent process of re-colonisation (*Chapman & Underwood, 1996*; *Smith & Otway, 1997*; *Liversage et al., 2012*). Similarly, when boulders are artificially added to a shoreline for habitat restoration, it is important to know how species such as chitons will colonise those boulders (*Chapman, 2012, 2013*). The present study suggests adult colonisation will occur most readily when the substratum among boulders is a rock-platform or other surfaces of

low complexity, which corresponds with patterns of colonisation of habitat patches predicted from known movement patterns of other intertidal molluscs (*Underwood & Chapman, 1989*; *Chapman & Underwood, 1994*; *Erlandsson, Kostylev & Williams, 1999*; *Underwood, 2004*).

*Smith & Otway (1997)* showed that *I. smaragdinus* is less sensitive to disturbance (boulder overturning) compared to other chiton species. Nevertheless, our results indicate it has a highly overdispersed distribution among boulders, similar to most other species of *Ischnochiton* (*Grayson & Chapman, 2004*; *Liversage & Benkendorff, 2013*). Adults of other species may disperse in similar ways to *I. smaragdinus*, but less frequently, explaining why no other chitons except *Callochiton crocinus* were seen during the present study.

## CONCLUSION

This study shows that dispersal by adults of an *Ischnochiton* species occurs via active benthic movement during nocturnal low tides, with the extent of dispersal dependent on the type of substratum. This provides information necessary to predict responses to changes in habitat and the potential to colonise new areas during habitat restoration (*Chapman, 2012, 2013*). The novel methods used in this study of image-adjusted time-lapse photography from above the water's surface will be useful in additional studies as there is increasing interest in evaluating movements of mobile intertidal invertebrates in natural (*Martinez et al., 2017*) and artificial (*Browne & Chapman, 2014*; *Evans et al., 2016*; *Firth et al., 2016*) rock-pools.

### Funding
This work was funded by a grant from the Nature Foundation SA. The funders had no role in study design, data collection and analysis, decision to publish, or preparation of the manuscript.

### Grant Disclosures
The following grant information was disclosed by the authors:
Nature Foundation SA.

### Competing Interests
The authors declare that they have no competing interests.

### Author Contributions
- Kiran Liversage conceived and designed the experiments, performed the experiments, analysed the data, contributed reagents/materials/analysis tools, wrote the paper, prepared figures and/or tables, reviewed drafts of the paper.
- Kirsten Benkendorff contributed reagents/materials/analysis tools, wrote the paper, reviewed drafts of the paper.

## Data Availability

The raw data has been provided as Supplemental Dataset Files.

## Supplemental Information

Supplemental information for this article can be found online at http://dx.doi.org/10.7717/peerj.4180#supplemental-information.

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
