# Peer review of "The first observations of Ischnochiton (Mollusca, Polyplacophora) movement behaviour, with comparison between habitats differing in complexity"

_PeerJ, doi:10.7717/peerj.4180_

## Round 0.1 · original submission · Minor Revisions

Congratulations on what is generally very positive reviewer feedback.

·

Basic reporting

Overall a good paper on a neglected subject.
I was not able to open the Excel supporting dataset.

Experimental design

These are cryptic animals that live under rocks (pebbles, cobble, boulders), usually ones at least partially buried in clean sand, and also on rock covered with a thin layer of sand. While it is commendable that the observations were done without disturbing the habitat (and therefore altering their behaviour), there is no indication of what proportion of the population actually come up on to the top of the substrate and therefore can be seen, compared with animals that stay below the sand surface.

Validity of the findings

There is mention of “drifting” dispersal as adults – this is a rare behaviour in chitons and is an escape response to disturbance, and is found commonly in only a few species – I. smaragdinus is not one of them. Species that use this live in mobile boulder fields usually in areas commonly subject to water movement sufficient enough to move the rocks. The vast majority of chitons respond to disturbance by clamping, then crawling into shadow. The very few species that use “drifting” react to disturbance by releasing from the substrate, curling up and falling down into crevices, then uncurling and reattaching – a useful survival behaviour if the substrate is badly disturbed. It is very obvious if this behaviour is exhibited by a species. Rather than just mention this as a possibility, it would have been very easy to see if this species exhibits this behaviour or not, and specify this.
They mention that this species has “an atypical carnivorous diet” - while the general opinion of chitons is that they are grazing herbivores, this is definitely not the case in Australian waters where grazing herbivores are the minority – this has been known since Kangas & Shepherd 1984, and is stated in the ABRS Fauna volume amongst other references. The commonest diet is grazing omnivores, grazing carnivores like I. smaragdinus are quite common.
Also, this is not the only species the emerge from the sand onto rock to graze at night, there are quite a few species that do this in SA, just not ones studied by Kangas & Shepherd.

·

Basic reporting

This is a well-written study showing basic movement patterns of a common intertidal chiton species and linking this to adult distribution.

Figure 1b is very blurry – do you have a better example?

Figure 2, 3 and 4 or their captions needs to specify sample (total number of boulders at each site for Fig 2, total number of chitons for Fig 3, total chitons measured for each metric in each substratum for Fig 4)

Experimental design

The experimental design is sound, with some careful considerations made by the authors (e.g. omitting data from chitons that interact with each other, inclusion of tortuosity of movement), but there are a few clarifications that should be made in the Methods as follows:

Clearly define substratum type categories and be consistent with terminology. You mention cobbles in the Results, but the description of the study area doesn’t define them. Clearly state how many substratum categories, their names and definition. If cobbles are not included in these, then remove mention of them from other section.

Line 122: specify that this task was undertaken during the day.

How did you take photos at night? Presumably with a flash? How do you control for possible effects on chiton?

Validity of the findings

The findings are well-linked to general patterns of adult distribution, with appropriate and cited speculation as to reasons for the results.

Line 260-261: how does this fit with other intertidal species’ known preferences?

Additional comments

Abstract: Please mention where this study took place (intertidal, South Australia)

Abstract, line 32: Specify the different habitat types in parentheses

Abstract, line 43: Please use ‘adult dispersal’ here to avoid confusion with larval ecology. You clarify this well in the ms, but the abstract stands alone.

Throughout ms: commas needed to separate independent clauses (e.g. Line 57, line 62). No commas needed for dependent clauses (line 79)

Line 277: specify what these ‘novel methods’ are.

·

Basic reporting

This is a concise and coherent manuscript that tells a reasonably simple story (based on only a few days of field sampling) but tells it well. It contains clearly articulated a priori hypotheses which are then adequately tested. As noted by the authors, there is not much published information on the ecology of underboulder chitons (though I am aware of some unpublished Masters theses on this topic), so there isn't much to directly compare their findings with. However, I did feel that a more in-depth consideration of the recorded movement patters of chitons from other habitats could have been included in the Discussion (see notes on the manuscript itself) .

Experimental design

There are 2 components to this study - the dispersion patterns of this chiton determined by overturning boulders/cobbles at 2 sites and counting the attached chitons underneath them; the movement patterns of chitons at one site recorded via time-lapse photography during night-time low tides. Both components have been adequately carried out, with appropriate collection of field data and statistical analyses of those data. The graphs and tables are well presented and clearly illustrate the main findings. There are a couple of minor queries on the ms itself, but no major improvements are required.

Validity of the findings

The data are robust, appropriately analysed and the conclusions generally sound. I did feel, however, that some commentary was required about possible movements of this species outside of the only situation that was investigated (ie night-time low tides) - what happens at high tide for example? There is not much speculation in the manuscript and when it occurs it is clearly identified as such. The authors quite rightly suggest that the technique they used has considerable merit for conducting further studies of movement/dispersion of cryptic species.

Additional comments

A simple yet illuminating study that provides important new information on a group of molluscs that has traditionally been hard to study. The material is worthy of publication.

---

## Round 0.2 · accepted · Accept

It is very pleasing to see more contributions on Australasian molluscan fauna in the literature.